# Locomotive Syndrome and Lumbar Spine Disease: A Systematic Review

**DOI:** 10.3390/jcm11051304

**Published:** 2022-02-27

**Authors:** Takaomi Kobayashi, Tadatsugu Morimoto, Koji Otani, Masaaki Mawatari

**Affiliations:** 1Department of Orthopaedic Surgery, Faculty of Medicine, Saga University, Saga 849-8501, Japan; takaomi_920@yahoo.co.jp (T.K.); mawatam@cc.saga-u.ac.jp (M.M.); 2Department of Orthopaedic Surgery, School of Medicine, Fukushima Medical University, Fukushima 960-1295, Japan; kotani@fmu.ac.jp

**Keywords:** locomotive syndrome, Loco-Check, 25-question Geriatric Locomotive Function Scale, 5-question Geriatric Locomotive Function Scale, Two-Step Test, Stand-Up Test, lumbar spine

## Abstract

Locomotive syndrome (LS) is defined based on the Loco-Check, 25-question Geriatric Locomotive Function Scale (GLFS-25), 5-question Geriatric Locomotive Function Scale (GLFS-5), Stand-Up Test, Two-Step Test, or a total assessment (i.e., positive for one or more of the GLFS-25, Stand-Up Test, and Two-Step Test). Lumbar spine disease has been reported to be one of the most common musculoskeletal disorders leading to LS. We therefore conducted a systematic review via PubMed, Google Scholar, Cochrane Library, Web of Science, and MEDLINE, based on the Preferred Reporting Items for Systematic Reviews and Meta-Analyses (PRISMA) guidelines. A total of 26 studies were considered to be eligible for inclusion in this systematic review. The GLFS-25 showed an association with low back pain, sagittal spinopelvic malalignment, and lumbar spinal stenosis but not vertebral fracture. The GLFS-5 showed an association with low back pain and lumbar spinal stenosis. The Loco-Check and Two-Step Test showed an association with low back pain, sagittal spinopelvic malalignment, and lumbar spinal stenosis. The Stand-Up Test showed no association with lumbar spinal stenosis. The total assessment showed an association with low back pain and lumbar spinal stenosis. Furthermore, the GLFS-25, Two-Step Test, and total assessment were improved by spinal surgery for lumbar spinal stenosis. The current evidence concerning the relationship between LS and lumbar spine disease still seems insufficient, so further investigations are required on this topic.

## 1. Introduction

The elderly population of Japan has continued to grow rapidly since the 1950s. According to the Ministry of Internal Affairs and Communications [1], people of ≥65 years of age numbered 4,110,000 in 1950 (5% of the population), 7,330,000 in 1970 (7% of the population), 14,930,000 in 1990 (12% of the population), 29,480,000 in 2010 (23% of the population), and 36,190,000 in 2020 (29% of the population). In 2007, the Japanese Orthopaedic Association (JOA) proposed the concept of locomotive syndrome (LS), which is defined as a state of reduced functional mobility due to musculoskeletal organ dysfunction [2,3]. LS occurs as the locomotive organs, such as bone (osteoporosis), joint and cartilage (osteoarthritis), muscle (sarcopenia), and intervertebral discs and nerves (spinal stenosis), deteriorate with aging [2,3]. 

The Loco-Check was designed as a simple self-administered check for LS that could be performed by individuals in the general population [2,3,4] (Figure 1). The Loco-Check includes seven items related to the activities of daily living (ADLs); the possible scores range from 0 to 7. Total scores of 0, 1, 2, and 3–7 points are to reflect non-LS, LS-1, LS-2, and LS-3, respectively. For healthcare professionals, the self-administered diagnostic tools for LS known as the 25-question Geriatric Locomotive Function Scale (GLFS-25) (Figure 2) and the 5-question Geriatric Locomotive Function Scale (GLFS-5) (Figure 3) were developed [5,6,7,8]. The GLFS-25 includes 25 items that are each graded on a 5-point scale (0–4 points) (possible scores range from 0 to 100). The domains covered by this scale include body pain (items 1–4), movement-related difficulty (items 5–7), usual care (items 8–11 and 14), social activities (items 12, 13, and 15–23), and cognition (items 24 and 25). Total scores of 0–6, 7–15, 16–23, and 24–100 are considered to reflect non-LS, LS-1, LS-2, and LS-3, respectively. The GLFS-5 is a 5-item version of the questionnaire and includes five items that are each graded on a 5-point scale (0–4 points) (possible scores range from 0 to 20). Total scores of 0–2, 3–5, 6–8, and 9–20 are considered to reflect non-LS, LS-1, LS-2, and LS-3, respectively. 

In addition, the JOA prescribed other official diagnostic tests, including the Stand-Up Test (Figure 4) and Two-Step Test (Figure 5) [9]. The Stand-Up Test evaluates lower limb strength according to stand—in a single-leg or double-leg stance— from four different heights (10, 20, 30, and 40 cm). The test is scored as 0–8, with the scores defined as follows: 0 (unable to stand); 1–4 (able to stand—using both legs—from 40, 30, 20, and 10 cm, respectively); and 5–8 (able to stand—using one leg—from 40, 30, 20, and 10 cm, respectively). Stand-Up Test scores of 0–1, 2, 3–4, and 5–8 points are equivalent to LS-3, LS-2, LS-1, and non-LS, respectively. The Two-Step Test evaluates walking ability. It is scored by normalizing the maximal length of two steps by the height. Two-Step Test scores <0.9, <1.1, <1.3, and ≥1.3 points correspond to LS-3, LS-2, LS-1 and non-LS, respectively. To prevent the demand for nursing care in the future, physical exercise is encouraged in patients with LS-1. To investigate musculoskeletal disorders that cause LS, orthopedic consultation is recommended for patients with LS-2. The utility of surgical intervention for LS-3 is an ongoing debate, but such an approach is thought to help improve the physical function.

In the outpatient department of orthopedics, lumbar spine disease has been reported as an extremely common musculoskeletal disorder leading to LS [10,11,12]. For instance, 64.6–80.6% of community dwelling residents were reported as diagnosed with lumbar spondylosis [11,12]. Furthermore, 10.7–17.6% of community dwelling residents were reported to suffer from associated symptoms [10]. Therefore, clarifying the relationship between LS and lumbar spine disease is an urgent issue. We conducted a systematic review on the relationship between LS and lumbar spine disease.

## 2. Materials and Methods

We conducted the present systematic review, based on the Preferred Reporting Items for Systematic Reviews and Meta-Analyses (PRISMA) statement [13]. All searches were conducted on 15 February 2022. We searched PubMed, Google Scholar, Cochrane Library, Web of Science, and MEDLINE for relevant English language peer-reviewed articles on the relationship between LS and lumbar spine disease. The following search phrase was used in PubMed: (locomotive syndrome [Title/Abstract]) AND (spine [Title/Abstract]). Other databases were carefully investigated by means of similar search strategies. Articles that were review articles, case reports (*n* < 3), commentary, editorial, insight articles, or proceedings were also reviewed. We excluded articles that did not mention the relationship between LS and lumbar spine disorders. According to previous reports [2,3,4,5,6,7,8,9], we defined LS according to the results of the Loco-Check, GLFS-25, GLFS-5, Stand-Up Test, Two-Step Test, or a total assessment (i.e., positive for one or more of the GLFS-25, Stand-Up Test, and Two-Step Test). We searched for unpublished or gray literature and screened websites, organizations, or reference lists of studies identified through the database search. Two researchers (T.K. and T.M.) independently assessed the paper selection. Any disagreements were discussed and resolved. The quality of the included studies was assessed based on the Newcastle–Ottawa Scale [14,15]. The following data were extracted: first author, publication year, study type, subject (i.e., number of patients, age, and sex), diagnostic test for LS, and clinical outcomes. Two researchers (T.K. and T.M.) independently assessed the quality of the included studies and extracted the data. Any disagreements were discussed and resolved.

## 3. Results

The initial database search identified 135 studies. After removing duplicates, 63 studies were screened. Finally, 26 studies (Table 1) were considered eligible for inclusion in this systematic review (Figure 6). The Newcastle–Ottawa Scale scores for the selected studies ranged from 5 to 9 (Table 2).

In the included studies, we found that lumbar spine disease included low back pain, vertebral fracture, sagittal spinopelvic malalignment, and lumbar spinal stenosis.

### 3.1. LS and Low Back Pain

Although low back pain is multifactorial, it is one of the most commonly encountered symptoms related to lumbar spine diseases in daily practice, accounting for 12.9–15.8% of cases [10]. Low back pain was strongly related to disc degeneration [42]. The presence of low back pain showed an association with the Loco-Check [16,42], GLFS-25 [18,19,20,21,22,23], GLFS-5 [24], Two-Step Test [25], and total assessment [26,27] (Table 1). Furthermore, the degree of low back pain showed a positive association with the GLFS-25 score [20,21,22,23] and a negative association with Two-Step Test score [25].

### 3.2. LS and Vertebral Fracture

The prevalence of vertebral fractures is 11.8–13.8% in the general population [43]. Nevertheless, only one paper had previously investigated the relationship between LS and vertebral fracture; Chiba et al. [28] reported that the GLFS-25 showed no association with the presence of vertebral fracture (Table 1).

### 3.3. LS and Sagittal Spinopelvic Malalignment

Various studies in the field of adult spinal deformity have described spinal sagittal imbalance as risk factor for a worsening in the quality of life [23,29,30,31,32,33,34,44,45,46]. Sagittal spinopelvic malalignment—flatback deformity (low pelvic tilt (PT), low sacral slope (SS), low lumbar lordosis (LL), and high pelvic incidence (PI)-LL mismatch) [46] and positive sagittal balance (high sagittal vertical axis (SVA), and high spinal inclination angle (SIA)) [47,48] (Figure 7)—showed an association with the Loco-Check [31,32,33,34], GLFS-25 [23,29,30,35,48], and Two-Step Test [25] (Table 1). There was no evidence concerning the relationship between the GLFS-5, Stand-Up Test, and total assessment and sagittal spinopelvic malalignment.

### 3.4. LS and Lumbar Spinal Stenosis

Lumbar spinal stenosis is the most commonly encountered lumbar spine disorders in daily practice, accounting for 10.7–12.9% of cases [10]. Lumbar spinal stenosis showed an association with the Loco-Check [36], GLFS-25 [28,37], GLFS-5 [24], Two-Step Test [25], and total assessment [39] but not the Stand-Up Test [39] (Table 1). Furthermore, spinal surgery for lumbar spinal stenosis (i.e., posterior decompression or short segment spinal fusion surgeries) improves the GLFS-25 [39,40,41,48], Two-Step Test [39,40,41], and total assessment [39,40,41], but not the Stand-Up Test [39,40,41] (Table 1). There was no report regarding the effect of surgery for lumbar spine disorders on the Loco-Check and GLFS-5.

## 4. Discussion

This systematic review describes the available evidence concerning the relationship between LS and lumbar spine disease. In the included studies, we found that lumbar spine disease included low back pain, vertebral fracture, sagittal spinopelvic malalignment, and lumbar spinal stenosis. Our findings were that LS showed an overall association with low back pain, sagittal spinopelvic malalignment, and lumbar spinal stenosis but not vertebral fracture.

### 4.1. LS and Low Back Pain

Our findings are influenced by the fact that the questionnaires include items concerning low back pain or its related quality of life problems [42] (Figure 1)—item 2 (pain in the back), 3 (pain in the leg), 4 (pain on moving), 5 (getting up and lying down), 6 (getting up from a chair), 7 (walking around the house), 9 (difficulty putting on trousers), 10 (difficulty using the toilet), 11 (difficulty bathing), 12 (going up and down stairs), 13 (walking briskly), 15 (walking continuously), 16 (going outside), 17 (carrying 2 kg), 20 (doing heavy housework), 21 (playing sports), 24 (anxious about falling), and 25 (anxious about walking)—and that the stride length was negatively correlated with the degree of low back pain [18,23,49,50]. There is no evidence concerning the relationship between the Stand-Up Test and low back pain. More specifically, previous studies found that low back pain affected sit-to-stand movement [51,52], but its association with LS remains unknown. Further investigations on this topic are considered necessary.

### 4.2. LS and Vertebral Fracture

The health-related quality of life, which is evaluated using the SF-12 Physical Component Summary score, back pain, and physical function assessed using the one-leg standing, timed up-and-go, walking speed, 30-s chair stand test, and maximum grip strength evaluations showed a significant association with both the severity and number of vertebral fractures in older women [53,54,55]. These findings suggest that vertebral fractures may affect GLFS-25. 

The inconsistency with our present findings can be explained by three reasons. First, the statistical method used is insufficient; previous authors analyzed this topic by gender, using the χ2 test despite the markedly low prevalence of vertebral fractures [28]. Second, the definition of vertebral fracture has varied among studies. Generally, ‘vertebral fracture’ was considered to be a compressive deformity wherein the height of the vertebra was >20% of the height of the adjacent uncompressed vertebra (20–25%, mild; >25–40%, moderate; >40%, severe) in lateral lumbar radiographs. Third, the relevance of our result regarding the relationship between LS and vertebral fracture is affected by the fact that only one retrospective study addressed this question. Further investigations are needed on the topic.

### 4.3. LS and Sagittal Spinopelvic Malalignment

Our results are consistent with previous reports, suggesting that spinopelvic malalignment may be a trigger for suspecting LS. Among the spinopelvic parameters, the SIA is reported to be the most relevant one for LS, and a SIA of ≥6° has a sensitivity of 52% and specificity of 87% for diagnosing LS-2 (GLFS-25 total score ≥16 points) [23].

The relationship between LS and lumbar flexibility remains unexplored. There is no evidence regarding the relationship between LS and the lumbo–pelvic complex [35,47,56]; individuals with a higher PI value are likely to have a higher LL value [35,47,56]. Furthermore, their lumbar facet joint is likely to have a more sagittal orientation [57,58], their lumbar facet joint contact force is likely to be lower in flexion–extension [59,60], and their anatomical acetabular anteversion angle is likely to be lower [35]. These conditions can be expected to lead individuals to greater use of their spine in ADLs (‘spine users’). Conversely, individuals with lower PI values can be expected to have lower LL values [35,47,56], their lumbar facet joint orientation is likely to be more coronal [57,58], their lumbar facet joint contact force in flexion–extension is likely to be higher [59,60], and their anatomical acetabular anteversion angle is likely to be higher [35]. These factors would lead to greater use of the hips in ADLs (‘hip users’). Further studies should be undertaken to validate these hypotheses.

### 4.4. LS and Lumbar Spinal Stenosis

Our findings may result from the hypothesis that the impairment of strength and balance in combination with gait disturbance with the pain status due to lumbar spinal stenosis affect scores of questionnaires (e.g., GLFS-25 item 3–25) and the Two-step Test [28,37,38,50]. In contrast, the Stand-Up Test predominantly measures the knee extension strength of the quadriceps femoris muscle, a parameter that stenosis does not directly affect at the most frequently responsible level (L4/5) [40,41].

Given the above, it is necessary to consider which LS test is the most useful in the field of spinal surgery. Preoperatively, the severity of LS based on the GLFS-25 and total assessment is almost the same [39,40,41]; in patients scheduled for primary surgery for the treatment of lumbar spinal stenosis, the prevalence of LS-1 and LS-2 assessed by GLFS-25 was 2.4–6.9% and 93.1–97.6%, respectively. Similarly, in patients scheduled to undergo primary surgery for lumbar spinal stenosis, the prevalence of LS-1 and LS-2 assessed by the total assessment were 1.8–5.0% and 95.0–98.2%, respectively. This indicates that the GLFS-25 is sufficient for the preoperative evaluation of severity of LS. Postoperatively, however, the GLFS-25 has a lower prevalence of LS-2 than the total assessment [39,40,41]. Among patients who received primary surgery for the treatment of lumbar spinal stenosis 1 year later, the prevalence of LS-1 and LS-2 assessed by GLFS-25 was 22.4–23.8% and 59.4–66.1%, respectively. In contrast, in patients who underwent primary surgery for lumbar spinal stenosis 1 year later, the prevalence of LS-1 and LS-2 assessed by the total assessment were 22.8–24.2% and 71.3–74.5%, respectively. This indicates that not only the GLFS-25 but also the total assessment (i.e., positive for one or more of the GLFS-25, Stand-Up Test, and Two-Step Test) is useful for accurately evaluating the improvement induced by spinal surgery. In other words, not only the GLFS-25 but also the Two-Step Test is useful for assessing the improvement induced by spinal surgery, as the Stand-Up Test showed no association with lumbar spinal stenosis [39]. Accordingly, the subjective and objective evaluations may differ, making it preferable to evaluate both in order to confirm the improvement induced by spinal surgery. 

Further studies are needed to investigate whether the surgical improvement of LS can extend the healthy life expectancy of individuals with spinal disorders.

### 4.5. Limitations of This Systematic Review

This systematic review is not homogenous and has several limitations. Firstly, and most importantly, the majority of reports on this topic are from Japan, which may have introduced some publication bias. The concept of LS is an important point in aging societies, so reports from countries other than Japan are expected to accumulate in the future. Secondly, an English language bias may exist, with important data published in Japanese possibly being omitted. Thirdly, the information in this systematic review was limited to the studies included. Therefore, lumbar spine disease included only low back pain, vertebral fracture, sagittal spinopelvic malalignment, and lumbar spinal stenosis. Furthermore, we could not quantitatively evaluate the combined data of the eligible studies due to differences in data quality and research design. Due to these limitations, we concluded that the current evidence is still insufficient. 

## 5. Conclusions

This systematic review described available evidence on the relationship between LS and lumbar spine disease (i.e., low back pain, vertebral fracture, sagittal spinopelvic malalignment, and lumbar spinal stenosis). The GLFS-25 showed an association with low back pain, sagittal spinopelvic malalignment, and lumbar spinal stenosis but not vertebral fracture. The GLFS-5 showed an association with low back pain and lumbar spinal stenosis. The Loco-Check and Two-Step Test showed an association with low back pain, sagittal spinopelvic malalignment, and lumbar spinal stenosis. The Stand-Up Test showed no association with lumbar spinal stenosis. The total assessment showed an association with low back pain and lumbar spinal stenosis. Furthermore, the GLFS-25, Two-Step Test, and total assessment were improved by spinal surgery for lumbar spinal stenosis

We delved into the detailed relationship between LS and lumbar spine disease via a systematic review and found that the current evidence was still insufficient to conduct a quantitative assessment. Further investigations are therefore warranted on this topic.

## Figures and Tables

**Figure 1 jcm-11-01304-f001:**
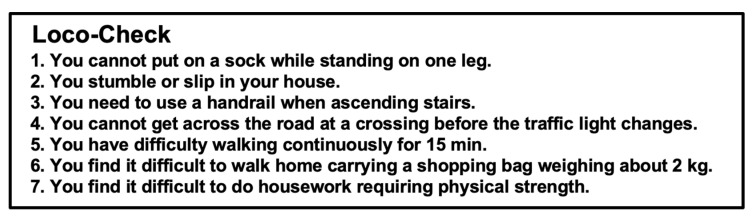
The Loco-Check. The Loco-Check includes seven items related to activities of daily living (ADLs); the possible scores range from 0 to 7. Total scores of 0, 1, 2, and 3–7 points are to reflect non-locomotive syndrome (LS), LS-1, LS-2, and LS-3, respectively.

**Figure 2 jcm-11-01304-f002:**
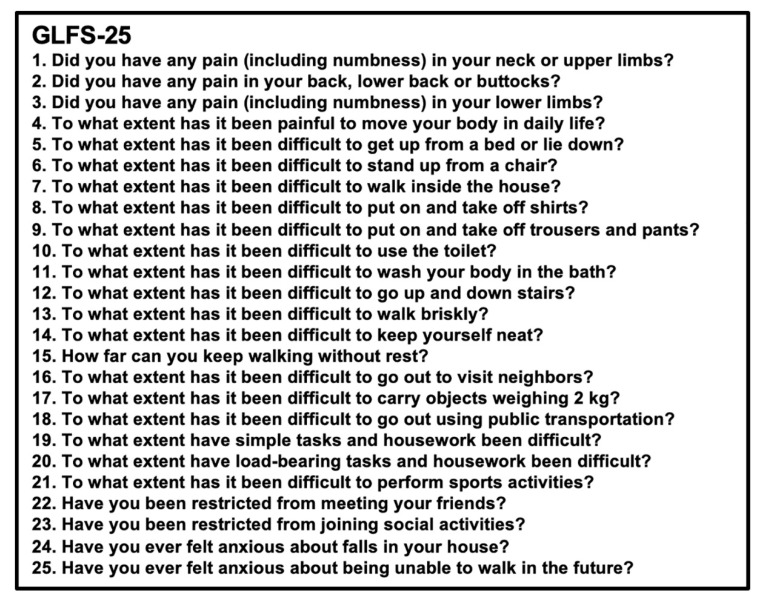
The 25-question Geriatric Locomotive Function Scale (GLFS-25). The GLFS-25 includes 25 items that are each graded on a 5-point scale (0–4 points) (possible scores range from 0 to 100). The domains covered by this scale include body pain (items 1–4), movement-related difficulty (items 5–7), usual care (items 8–11 and 14), social activities (items 12, 13, and 15–23), and cognition (items 24 and 25). Total scores of 0–6, 7–15, 16–23, and 24–100 are considered to reflect non-LS, LS-1, LS-2, and LS-3, respectively.

**Figure 3 jcm-11-01304-f003:**
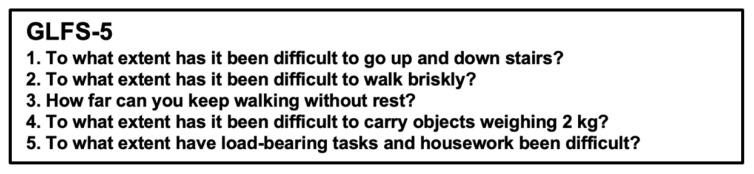
The 5-question Geriatric Locomotive Function Scale (GLFS-5). The GLFS-5 is a 5-item version of the questionnaire and includes five items that are each graded on a 5-point scale (0–4 points) (possible scores range from 0 to 20). Total scores of 0–2, 3–5, 6–8, and 9–20 are considered to reflect non-locomotive syndrome (LS), LS-1, LS-2, and LS-3, respectively.

**Figure 4 jcm-11-01304-f004:**
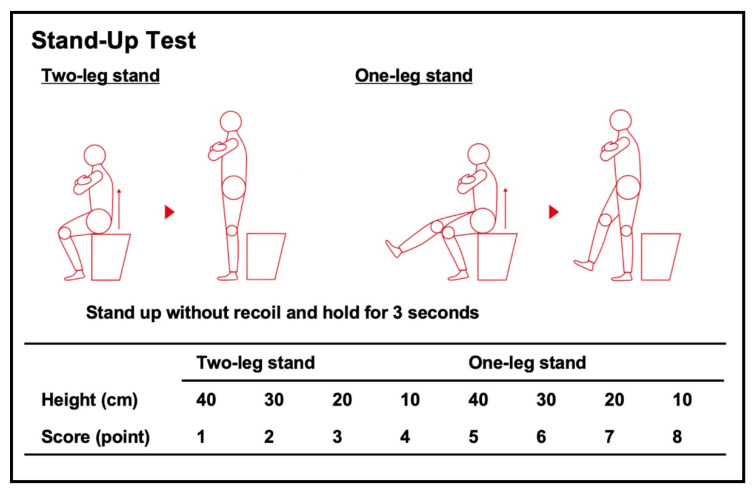
The Stand-Up Test. The Stand-Up Test evaluates lower limb strength according to stand—in a single-leg or double-leg stance—from 4 different heights (10, 20, 30, and 40 cm). The test is scored as 0–8, with the scores defined as follows: 0 (unable to stand); 1–4 (able to stand—using both legs—from 40, 30, 20, and 10 cm, respectively); and 5–8 (able to stand—using one leg—from 40, 30, 20, and 10 cm, respectively). Stand-Up Test scores of 0–1, 2, 3–4, and 5–8 points are equivalent to LS-3, LS-2, LS-1, and non-LS, respectively. The reproduction of this figure is permitted by the Japanese Orthopaedic Association (JOA) locomotive syndrome prevention awareness official website [9].

**Figure 5 jcm-11-01304-f005:**
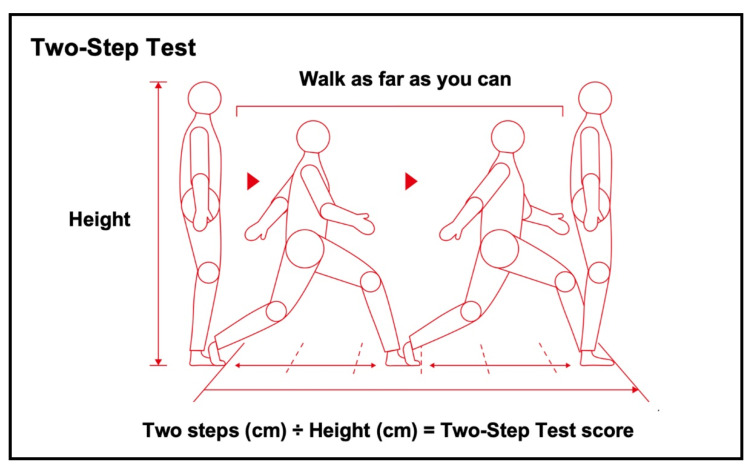
The Two-Step Test. The Two-Step Test evaluates walking ability. It is scored by normalizing the maximal length of two steps by the height. Two-Step Test scores <0.9, <1.1, <1.3, and ≥1.3 points correspond to LS-3, LS-2, LS-1, and non-LS, respectively. The reproduction of this figure is permitted by the Japanese Orthopaedic Association (JOA) locomotive syndrome prevention awareness official website [9].

**Figure 6 jcm-11-01304-f006:**
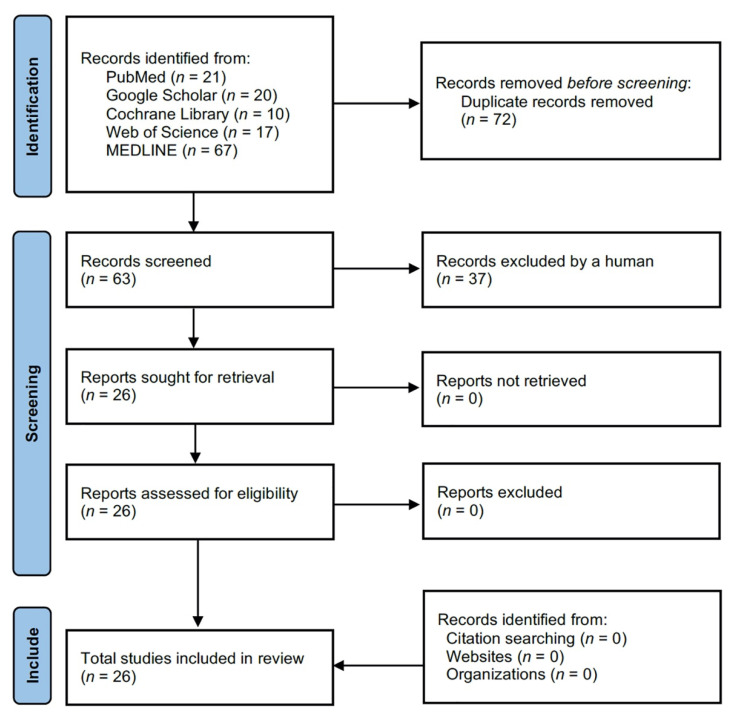
The Preferred Reporting Items for Systematic Reviews and Meta-Analyses (PRIZMA) [13] flow chart of the paper selection.

**Figure 7 jcm-11-01304-f007:**
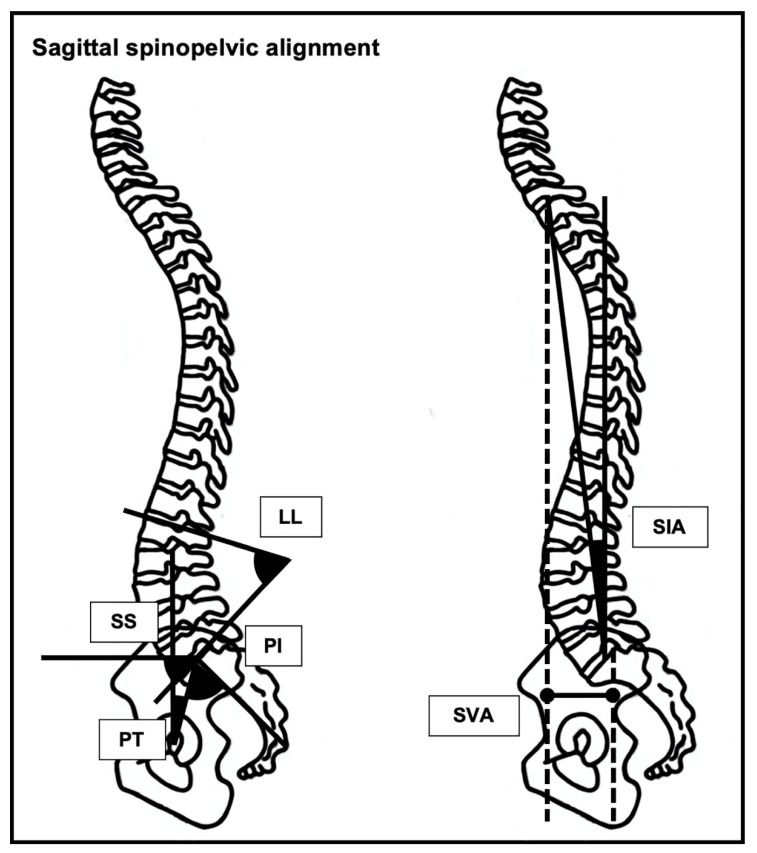
Sagittal spinopelvic alignment includes the pelvic incidence (PI), pelvic tilt (PT), sacral slope (SS), lumbar lordosis (LL), sagittal vertical axis (SVA), and spinal inclination angle (SIA). The PI is the angle between a line perpendicular to the sacral plate at its midpoint and a line connecting this point to the bi-coxo-femoral axis. The PT is the angle between a vertical line passing through the bi-coxo-femoral axis and a line joining the bi-coxo-femoral axis with the center of the upper sacral endplate. The SS is the angle between a tangent line to the superior endplate of S1 and the horizontal plane. The LL is the angle between the superior endplate of L1 and the upper sacral endplate. The SVA is the horizontal distance between a plumb line drawn from the center of C7 and a line drawn from the center of C7 to the posterior superior corner of S1. The SIA is the angle between the true vertical and a straight line from the tip of the T1 spinous process to that of S1.

**Table 1 jcm-11-01304-t001:** Summary of the results.

Study	Design	Subject	LS	Outcome
Kasukawa et al., 2020 [16]	Cross-sectional study	253 healthy volunteers (118 men, 135 women), age 60–88 years	Loco-Check	Low back pain
Sasaki et al., 2013 [17]	Cross-sectional study	727 healthy volunteers (264 men, 463 women), age 56.6 ± 13.6 (21–87) years	Loco-Check	Low back pain
Iizuka et al., 2015 [18]	Cross-sectional study	287 healthy volunteers (100 men, 187 women), age 64.7 ± 11.2 (40–89) years	GLFS-25	Low back pain
Taniguchi et al., 2021 [19]	Cross-sectional study	2077 healthy volunteers (730 men, 1347 women), age 68.3 ± 5.4 (30–74) years	GLFS-25	Low back pain
Muramoto et al., 2012 [20]	Cross-sectional study	358 healthy volunteers (128 men, 230 women), age 66.0 ± 10.0 (40–91) years	GLFS-25	Low back pain
Muramoto et al., 2013 [21]	Cross-sectional study	406 healthy volunteers (167 men, 239 women), age 68.8 ± 6.7 (60–88) years	GLFS-25	Low back pain
Muramoto et al., 2014 [22]	Cross-sectional study	217 healthy volunteers (217 women), age 68.2 ± 5.0 (60–79) years	GLFS-25	Low back pain
Muramoto et al., 2016 [23]	Cross-sectional study	125 healthy volunteers (125 women), age 66.2 ± 9.7 (40–88) years	GLFS-25	Low back pain, sagittal spinopelvic alignment
Matsumoto et al., 2016 [24]	Cross-sectional study	223 healthy volunteers (82 men, 141 women), age 73.6 ± 8.3 years	GLFS-5	Low back pain, lumbar spinal stenosis
Fujita et al., 2019 [25]	Cross-sectional study	357 patients scheduled to undergo primary surgery for lumbar spinal stenosis (201 men, 156 women), 73.3 ± 5.5 years	Two-Step Test	Low back pain, sagittal spinopelvic alignment, lumbar spinal stenosis
Imagama 2017 [26]	Cross-sectional study	523 healthy volunteers (240 men, 283 women), age 63.3 ± 10.0 years	Total assessment	Low back pain
Nishimura 2020 [27]	Cross-sectional study	715 workers (579 men, 136 women), age 44.6 ± 10.0 (18–64) years	Total assessment	Low back pain
Chiba et al., 2016 [28]	Cross-sectional study	647 healthy volunteers (247 men, 400 women), age 58.4 ± 11.0 years	GLFS-25	Vertebral fracture, lumbar spinal stenosis
Machino et al., 2020 [29]	Cross-sectional study	211 healthy volunteers (89 men, 122 women), age 64.0 ± 10.1 years	GLFS-25	Sagittal spinopelvic alignment
Machino et al., 2020 [30]	Cross-sectional study	448 healthy volunteers (184 men, 264 women), age 62.7 years	GLFS-25	Sagittal spinopelvic alignment
Hirano et al., 2012 [31]	Cross-sectional study	386 healthy volunteers (131 men, 233 women), age 67.6 ± 8.7 (50–91) years	Loco-Check	Sagittal spinopelvic alignment
Hirano et al., 2012 [32]	Cross-sectional study	135 healthy volunteers (54 men, 81 women), 76.5 ± 4.7 (70–90) years	Loco-Check	Sagittal spinopelvic alignment
Hirano et al., 2013 [33]	Cross-sectional study	187 healthy volunteers (187 women), age 68.0 ± 8.3 years	Loco-Check	Sagittal spinopelvic alignment
Hirano et al., 2012 [34]	Cross-sectional study	105 healthy volunteers (105 men), age 69.5 ± 8.2 (50–90) years	Loco-Check	Sagittal spinopelvic alignment
Ohba et al., 2021 [35]	Retrospective cohort study	40 patients with a diagnosis of adult spinal deformity who underwent spinal surgery (3 men, 37 women), age 72.6 ± 5.9 years	GLFS-25	Sagittal spinopelvic alignment
Shigematsu et al., 2019 [36]	Case–control study	28 patients with lumbar spinal stenosis who underwent spinal surgery (15 men, 13 women), age 73.7 ± 5.6 years 46 elderly persons (16 men, 30 women), age 73.9 ± 5.4 years	Loco-Check	Lumbar spinal stenosis
Araki et al., 2021 [37]	Cross-sectional study	82 patients with lumbar spinal stenosis who underwent decompression surgery (47 men, 35 women), age 73.4 ± 8.4 years	GLFS-25	Lumbar spinal stenosis
Fujita et al., 2019 [38]	Cross-sectional study	200 patients scheduled to undergo primary surgery for lumbar spinal stenosis (120 men, 80 women), age 73.2 years	Total assessment	Lumbar spinal stenosis
Shimizu et al., 2021 [39]	Prospective cohort study	101 patients scheduled to undergo primary surgery for lumbar spinal stenosis (46 men, 55 women), age 69.3 ± 8.1 years	Total assessment	Surgery for lumbar spinal stenosis
Kato et al., 2020 [40]	Prospective cohort study	257 patients who underwent surgery for degenerative diseases of the lumbar spine (209 men, 48 women), age 71.5 ± 6.9 years	Total assessment	Surgery for lumbar spinal stenosis
Fujita et al., 2020 [41]	Prospective cohort study	166 patients scheduled to undergo primary surgery for lumbar spinal stenosis (95 men, 71 women), age 72.8 ± 5.5 years	Total assessment	Surgery for lumbar spinal stenosis

LS: locomotive syndrome; GLFS-25: the 25-question Geriatric Locomotive Function Scale; GLFS-5: the 5-question Geriatric Locomotive Function Scale.

**Table 2 jcm-11-01304-t002:** A quality assessment of the eligible studies based on the Newcastle–Ottawa Scale [14,15].

Study	Selection	Comparability	Outcome/Exposure	Total Score
Kasukawa et al., 2020 [16]	★★★★		★★	6
Sasaki et al., 2013 [17]	★★★★	★★	★★	8
Iizuka et al., 2015 [18]	★★★★	★★	★★	8
Taniguchi et al., 2021 [19]	★★★★		★★	6
Muramoto et al., 2012 [20]	★★★★	★★	★★	8
Muramoto et al., 2013 [21]	★★★	★★	★★	7
Muramoto et al., 2014 [22]	★★	★★	★★	6
Muramoto et al., 2016 [23]	★★	★★	★★	6
Matsumoto et al., 2016 [24]	★★★★		★★	6
Fujita et al., 2019 [25]	★★★★★	★★	★★	9
Imagama 2017 [26]	★★★★		★★★	7
Nishimura 2020 [27]	★★★		★★	5
Chiba et al., 2016 [28]	★★★	★★	★★	7
Machino et al., 2020 [29]	★★★★		★★★	7
Machino et al., 2020 [30]	★★★★★		★★★	8
Hirano et al., 2013 [31]	★★★★		★★★	7
Hirano et al., 2012 [32]	★★★★	★★	★★★	9
Hirano et al., 2013 [33]	★★★	★★	★★★	8
Hirano et al., 2012 [34]	★★★	★★	★★★	8
Ohba et al., 2021 [35]	★★★★	★★	★★★	9
Shigematsu et al., 2019 [36]	★★★	★★	★★	7
Araki et al., 2021 [37]	★★★		★★	5
Fujita et al., 2019 [38]	★★★★		★★★	7
Shimizu et al., 2021 [39]	★★★★	★★	★★	8
Kato et al., 2020 [40]	★★★★	★★	★★	8
Fujita et al., 2020 [41]	★★★★	★★	★★	8

Newcastle-Ottawa Scale for case-control studies: Selection (Maximum ★★★★)—(1) Is the case definition adequate? (2) Representativeness of the cases; (3) Selection of controls; (4) Definition of Controls. Comparability (Maximum ★★) 2013 (1) Confounding factors controlled. Exposure (Maximum ★★★)—(1) Ascertainment of exposure; (2) Same method of ascertainment for cases and controls; (3) Non-Response rate. Newcastle-Ottawa Scale for cohort studies: Selection (Maximum ★★★★)—(1) Representativeness of the exposed cohort; (2) Selection of the non-exposed cohort; (3) Ascertainment of exposure; (4) Demonstration that outcome of interest was not present at start of study. Comparability (Maximum ★★)—(1) Confounding factors controlled. Outcome (Maximum ★★★)—(1) Assessment of outcome; (2) Was follow-up long enough for outcomes to occur; (3) Adequacy of follow up of cohorts. Newcastle-Ottawa Scale adapted for cross-sectional studies: Selection (Maximum ★★★★★)—(1) Representativeness of the sample; (2) Sample size; (3) Non-respondents; (4) Ascertainment of the exposure. Comparability: (Maximum ★★)—(1) Confounding factors controlled. Outcome (Maximum ★★★)—(1) Assessment of outcome; (2) Statistical test.

## Data Availability

Not applicable.

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
