# Peer review of "Locomotive Syndrome and Lumbar Spine Disease: A Systematic Review"

_jcm, 2022, doi:10.3390/jcm11051304_

Round 1

Reviewer 1 Report

The aim of the authors was to demonstrated the relationship between the Locomotive Syndrome (LS) and lumbar spine diseases.

In the first part of the manuscript, they presented the LS with an extensive description. This syndrome is well known in the Japanese papulation and based on different Tests like Loco-Check, 25-question Geriatric Locomotive Function Scale (GLFS-25), 5-question Geriatric Locomotive Scale Function Scale (GLFS-5), Stand-Up Test, Two-Step Test. All of these tests were descripted and illustrated by the authors.

In the second part the authors presented their searching for relevant peer-reviewed article on PubMed, Google, Scholar, Cochrane Library, Web of Science, and MEDLINE which reported about LS and lumbar spine diseases. The author pointed out that they did not perform a systematic review (page 6, line 110).

The authors reported the limitations of their study honestly. The main limitation of this study is the publication bias in my opinion. The majority of the included publications are from Japan.

The manuscript is full of expectations theories, without scientific background. The results and conclusion are empty generalities.

Finally, although the article is well written, with an interesting topic for European population. Nevertheless, I´m sorry to say, that the conclusion cannot be supported from the data sufficiently. In my opinion this manuscript could be published only with an own patient series including a systematic review.

Author Response

We are grateful to Reviewer #1 for the useful suggestions which have greatly helped us to improve our paper. As indicated in the responses that follow, we have taken all of these comments and suggestions into account in our revision of the manuscript.

Comment 1: The aim of the authors was to demonstrated the relationship between the Locomotive Syndrome (LS) and lumbar spine diseases. In the first part of the manuscript, they presented the LS with an extensive description. This syndrome is well known in the Japanese papulation and based on different Tests like Loco-Check, 25-question Geriatric Locomotive Function Scale (GLFS-25), 5-question Geriatric Locomotive Scale Function Scale (GLFS-5), Stand-Up Test, Two-Step Test. All of these tests were descripted and illustrated by the authors.

Response 1: Thank you very much for your comments. We have taken all of these comments and suggestions into account in our revision of the manuscript.

Comment 2: In the second part the authors presented their searching for relevant peer-reviewed article on PubMed, Google, Scholar, Cochrane Library, Web of Science, and MEDLINE which reported about LS and lumbar spine diseases. The author pointed out that they did not perform a systematic review (page 6, line 110).

Response 2: According to your suggestion, we conducted a systematic review.

MANUSCRIPT:

We conducted a systematic review on the relationship between LS and lumbar spine disease. (Page 2, Line 115–116)

Comment 3: The authors reported the limitations of their study honestly. The main limitation of this study is the publication bias in my opinion. The majority of the included publications are from Japan.

Response 3: Thank you for your insightful indication. We completely agree with you. We have added the limitation.

MANUSCRIPT:

Firstly, and most importantly, the majority of reports on this topic are from Japan, which may have introduced some publication bias. (Page 11, Line 631–633)

Comment 4: The manuscript is full of expectations theories, without scientific background. The results and conclusion are empty generalities.

Response 4: According to your suggestion, we conducted a systematic review. Furthermore, we have added the limitation.

MANUSCRIPT:

We conducted a systematic review on the relationship between LS and lumbar spine disease. (Page 2, Line 115–116)

Furthermore, we could not make quantitative evaluate the combined data of the eligible studies due to differences in data quality and research design. Due to these limitations, we concluded that the current evidence is still insufficient. (Page 12, Line 708–710)

We delved into the detailed relationship between LS and lumbar spine disease via a systematic review and found that the current evidence was still insufficient to conduct a quantitative assessment. Further investigations are therefore warranted on this topic. (Page 12, Line 722–724)

Comment 5: Finally, although the article is well written, with an interesting topic for European population. Nevertheless, I´m sorry to say, that the conclusion cannot be supported from the data sufficiently. In my opinion this manuscript could be published only with an own patient series including a systematic review.

Response 5: Thank you for your indication. According to your suggestion, we conducted a systematic review.

MANUSCRIPT:

We conducted the present systematic review, based on the Preferred Reporting Items for Systematic Reviews and Meta-Analyses (PRISMA) statement [13]. (Page 3, Line 190–191)

We searched for unpublished or gray literature and screened websites, organizations, or reference lists of studies identified through the database search. Two researchers (T.K. and T.M.) independently assessed the paper selection. Any disagreements were discussed and resolved. The quality of the included studies was assessed based on the Newcastle–Ottawa Scale [14,15]. The following data were extracted: first author, publication year, study type, subject (i.e., number of patients, age, and sex), diagnostic test for LS, and clinical outcomes. Two researchers (T.K. and T.M.) independently assessed the quality of the included studies and extracted the data. Any disagreements were discussed and resolved. (Page 4, Line 222–230)

The initial database search identified 135 studies. After removing duplicates, 63 studies were screened. Finally, 26 studies (Table 1) were considered eligible for inclusion in this systematic review (Figure 6). The Newcastle–Ottawa Scale scores for the selected studies ranged from 5 to 9 (Table 2). (Page 4, Line 232–235, see figure 6)

This systematic review describes the available evidence concerning the relationship between LS and lumbar spine disease. (Page 10, Line 515–516)

REFERENCES:

  1. Page, M.J.; McKenzie, J.E.; Bossuyt, P.M.; Boutron, I.; Hoffmann, T.C.; Mulrow, C.D.; Shamseer, L.; Tetzlaff, 750 J.M.; Akl, E.A.; Brennan, S.E.; et al. The PRISMA 2020 statement: An updated guideline for reporting systematic reviews. B.M.J. 2021, 29, 372:n71.
  2. Stang, A. Critical evaluation of the Newcastle-Ottawa scale for the assessment of the quality of nonrandomized studies in meta-analyses. Eur. J. Epidemiol. 2010, 25, 603–605.
  3. Herzog, R.; Álvarez-Pasquin, M.J.; Díaz, C.; Del Barrio, J.L.; Estrada, J.M.; Gil, Á. Are healthcare workers' intentions to vaccinate related to their knowledge, beliefs and attitudes? A systematic review. BMC Public Health. 2013, 13, 154.

Reviewer 2 Report

The Authors provides a narrative review about the relationship between the locomotive syndrome and lumbar spine diseases. The paper is well written and clear exposed. The issue is of interest to the scientific community in the field of vertebral evaluation. The LS as a concept is used currently mainly in Japan, in the rest of the world it is spreading. With the aging of the population the LS should  surely be more explained and researched thoroughly. The manuscript represents a summary of the current knowledge of scales for vertebral and LS assessment.

Author Response

We are grateful to Reviewer #2 for the useful suggestions which have greatly helped us to improve our paper. As indicated in the responses that follow, we have taken all of these comments and suggestions into account in our revision of the manuscript.

Comment 1: The Authors provides a narrative review about the relationship between the locomotive syndrome and lumbar spine diseases. The paper is well written and clear exposed. The issue is of interest to the scientific community in the field of vertebral evaluation. The LS as a concept is used currently mainly in Japan, in the rest of the world it is spreading. With the aging of the population the LS should surely be more explained and researched thoroughly. The manuscript represents a summary of the current knowledge of scales for vertebral and LS assessment.

Response 1: Thank you for your review and comment. We have improved our manuscript. We hope that the revised version of our paper will now be considered suitable for publication in Journal of Clinical Medicine.

Reviewer 3 Report

I thank you for the great opportunity to read this article.

 General statement and summary

I have read with great interest and attention this article entitled "Locomotive syndrome and lumbar Spine disease: A Narrative review", submitted for publication in the journal JCM. In this article, the authors provide a systematic review of articles from PubMed and Google Academy, Cochrane Library, Web Sciences, and Medline, where a search was conducted for relevant articles published in English on the relationship between drugs and lumbar spine disease. This problem is very relevant and after a large literary review, the available evidence is still insufficient.

Specific Comments.

In order to improve the perception of the publication, it is necessary to make a number of changes:

  1. In the "Introduction" section, the authors need to add epidemiological data with references to the literature, this significantly objectifies the relevance of this problem.

There are no fundamental errors in the article. The work corresponds to the subject of the journal.

Author Response

We are grateful to Reviewer #3 for the useful suggestions which have greatly helped us to improve our paper. As indicated in the responses that follow, we have taken all of these comments and suggestions into account in our revision of the manuscript.

Comment 1: I have read with great interest and attention this article entitled "Locomotive syndrome and lumbar Spine disease: A Narrative review", submitted for publication in the journal JCM. In this article, the authors provide a systematic review of articles from PubMed and Google Academy, Cochrane Library, Web Sciences, and Medline, where a search was conducted for relevant articles published in English on the relationship between drugs and lumbar spine disease. This problem is very relevant and after a large literary review, the available evidence is still insufficient.

Response 1: Thank you very much for your comments. We have taken all of these comments and suggestions into account in our revision of the manuscript.

Comment 2: In order to improve the perception of the publication, it is necessary to make a number of changes: In the "Introduction" section, the authors need to add epidemiological data with references to the literature, this significantly objectifies the relevance of this problem.

Response 2: We completely agree with you. We have added epidemiological data with the relevant references.

MANUSCRIPT:

The elderly population of Japan has continued to grow rapidly since the 1950s. According to the Ministry of Internal Affairs and Communications [1], people of ≥65 years of age numbered 4,110,000 in 1950 (5% of the population), 7,330,000 in 1970 (7% of the population), 14,930,000 in 1990 (12% of the population), 29,480,000 in 2010 (23% of the population), and 36,190,000 in 2020 (29% of the population). (Page 1, Line 29–33)

For instance, 64.6–80.6% of community dwelling residents were reported to diagnosed with lumbar spondylosis [11,12]. Furthermore, 10.7–17.6% of community dwelling residents were reported to suffer from associated symptoms [10]. (Page 2, Line 111–114)

REFERENCES:

  1. The Ministry of Internal Affairs and Communications. https://www.mhlw.go.jp/toukei/list/81-1a.html. [Accessed 15 February 2022].
  2. Yoshimura, N.; Muraki, S.; Oka, H.; Mabuchi, A.; En-Yo, Y.; Yoshida, M.; Saika, A.; Yoshida, H.; Suzuki, T.; Yamamoto, S.; et al. Prevalence of knee osteoarthritis, lumbar spondylosis, and osteoporosis in Japanese men and women: The research on osteoarthritis/osteoporosis against disability study. J. Bone Miner. Metab. 2009, 27, 620–628.
  3. Yoshimura, N.; Muraki, S.; Nakamura, K.; Tanaka, S. Epidemiology of the locomotive syndrome: The research on osteoarthritis/osteoporosis against disability study 2005-2015. Mod. Rheumatol. 2017, 27, 1–7.

Comment 3: There are no fundamental errors in the article. The work corresponds to the subject of the journal.

Response 3: Thank you very much for your comments. We hope that the revised version of our paper will now be considered suitable for publication in Journal of Clinical Medicine.